# Grape Pomace Rich-Phenolics and Anthocyanins Extract: Production by Pressurized Liquid Extraction in Intermittent Process and Encapsulation by Spray-Drying

**DOI:** 10.3390/foods13020279

**Published:** 2024-01-16

**Authors:** Jessica Thaís do Prado Silva, Millene Henrique Borges, Carlos Antonio Cardoso de Souza, Carmen Sílvia Fávaro-Trindade, Paulo José do Amaral Sobral, Alessandra Lopes de Oliveira, Milena Martelli-Tosi

**Affiliations:** 1Faculty of Animal Science and Food Engineering, University of São Paulo, Av. Duque de Caxias Norte, 225, Pirassununga 13635-900, SP, Brazil; jessicatps@usp.br (J.T.d.P.S.); millenehenriqueborges@usp.br (M.H.B.); carlos.antonio.souza@usp.br (C.A.C.d.S.); carmenft@usp.br (C.S.F.-T.); pjsobral@usp.br (P.J.d.A.S.); alelopes@usp.br (A.L.d.O.); 2Postgraduate Programme in Materials Science and Engineering, Faculty of Animal Science and Food Engineering, University of São Paulo, Av. Duque de Caxias Norte, 225, Pirassununga 13635-900, SP, Brazil; 3Food Research Center (FoRC), Rua do Lago, 250, Semi-Industrial Building, Block C, São Paulo 05508-080, SP, Brazil

**Keywords:** agro-industrial residue, drying, microencapsulation, anthocyanins

## Abstract

A considerable number of grape pomaces are generated annually. It represents a rich source of bioactive compounds, such as phenolic compounds and anthocyanins. Pressurized liquid extraction (PLE) has emerged as a green technology for recovering bioactive compounds from vegetal matrixes. In our study, PLE parameters (temperature, number of cycles, and rinse volume) have been studied to produce grape pomace extracts with high bioactive content using an experimental design. The experimental data obtained were adjusted to linear and quadratic models. The first-order model was better in predicting anthocyanins contents (TA, R^2^ = 0.94), whereas the second-order model was predictive for total phenolic compounds (TPC, R^2^ = 0.96). The main process parameter for the recovery of bioactive compounds was temperature, and the results showed opposing behaviors concerning TPC and TA, as it is difficult to optimize conditions for both. The extract containing the higher concentration of TPC (97.4 ± 1.1 mg GAE/g, d.b.) was encapsulated by spray-drying using maltodextrin as wall material. Particles presented with a spherical shape (~7.73 ± 0.95 μm) with a recovery yield of 79%. The results demonstrated that extraction followed by encapsulation of grape pomace extract is a good strategy to simplify future applications, whether for food, cosmetics or pharmaceutical fields.

## 1. Introduction

Grape pomace is the most abundant solid by-product generated during winemaking. It is composed of skins, seeds, and any other solids remaining after pressing. It is a complex material, whose chemical composition can vary according to the winemaking’s operational conditions, equipment, and grape variety; however, it usually has polysaccharides as the main component (43–75%) and also contains proteins (6–15%), lipids, and a wide diversity of phenolic compounds [1].

The valorization of grape pomace is usually linked to its use as substrate for ethanol production in distilleries or as organic fertilizer. Nonetheless, the polyphenol fraction of grape pomace, which contains high value-added compounds, is unutilized in these applications. In fact, grape pomace from wine industries retains a large content of phenolic compounds, since their extraction from grape matrix is not complete during the vinification process. Thus, the extraction of phenolic compounds from grape pomace is a strategy to recover such valuable compounds for further useful application [2].

Solid–liquid extraction using water, acetone, methanol or ethanol is the conventional approach applied to extract phenolic compounds from grape pomace. In spite of being simple, such a technique requires a long extraction time, which could cause the degradation of phenolic compounds over time. Furthermore, in most cases, solid–liquid extraction cannot be considered a green alternative due to the high solvent consumption. Thus, the study of new approaches to extract phenolic compounds from grape pomace is encouraged.

The pressurized liquid extraction (PLE) that operates in an intermittent process was developed in the 1990s by Dionex as equipment for sample preparation [3]. PLE consists of a solid–liquid extraction operating at high temperatures and pressures (normally 10 MPa in the intermittent process) and reduced times. The high pressure keeps the solvent as a liquid, even at temperatures above its boiling point, and forces the solvent to penetrate through the pores of the matrix. High temperature reduces the viscosity and increases the diffusivity of the solvent, which allows for greater diffusion in the solid matrix. These two process variables (pressure and temperature) at high values promote higher mass transfer rates. As a consequence, the extraction process is faster and requires smaller amounts of solvents [4].

The PLE in intermittent process also demonstrates other advantages over conventional extraction because, in this process, a pressurized solvent passes through the extraction matrix under an inert atmosphere. This protects the compounds from degradation over time. Also, PLE in an intermittent process requires only small amounts of solvent and less time of operation, so it is considered a green extraction method [5].

Considering the sustainability aspects and efficiency of this extraction technique, our research group has studied the optimization of extraction with the perspective of industrial use [4,6,7,8,9,10].

The application of PLE has been shown to be a viable technique for the extraction of phenolic compounds from grape pomace [11,12]; however, process parameters must be carefully considered in order to increase the extraction of compounds and pigments with active properties to avoid their degradation. Moreover, the study must establish operational parameters that enable efficient extraction using GRAS solvents (Generally Recognized as Safe) that are sustainable and that enable extracts free of residues of toxic organic solvents that can be used in the food’s formulation.

Anthocyanins are the main class of the phenolic compounds present in grape pomace extract. They are composed of two aromatic rings joined with a heterocycle, thus being part of the flavonoid family [2]. Such compounds are responsible for providing color to grapes, making them interesting options to be used as water-soluble colorants for food, cosmetics and medicines, but notably, the stability of these molecules may be affected by several factors, such as pH, high temperatures of processing, high storage temperatures, light, oxygen, solvents, and metal ions. Given these factors, the study of anthocyanin-rich extract stabilization by spray-drying has been the main focus of many studies due to their high potential for use as a colorant, in addition to their beneficial effects [13,14,15,16,17,18,19,20,21].

Spray-drying is a continuous, simple and fast process in which a liquid is transformed into a powder. This process involves the nebulization of the liquid into a chamber that also receives a flow of hot air that promotes the instantaneous water evaporation of the droplets, turning them into particles. Therefore, this technique enables drying of heat-sensitive compounds (such as phenolic compounds) and producing matrix-type particles, where these compounds are involved and protected by the carrier used constituting an encapsulation technique.

In this context, the aims of this work were to study the production of phenolics-rich extracts using pressurized liquid extraction, to encapsulate the extract obtained from the best condition by spray-drying and to characterize the microparticles obtained regarding to morphological and chemical aspects (size distribution, Scanning Electron Microscopy and Fourier-Transform Infrared spectroscopy).

## 2. Materials and Methods

### 2.1. Materials

By-product from Bordeaux (*Vitis labrusca*) grape vinification was kindly provided by Adega Família Ferragut (Vinhedo, SP, Brazil). The by-product (grape peel and seeds) was collected after the pressing step of red wine production, then ground using a blender and stored at −20 °C. Maltodextrin (Mor-Rex 1910) was kindly donated by Ingredion (Mogi Guaçu, São Paulo, Brazil). Gallic acid, iron sulfate (FeSO_4_), and 2,4,6-Tris(2-pyridyl)-s-triazine (TPTZ) were purchased from Sigma Aldrich. All other chemicals were of analytical grade.

### 2.2. Pigment Extraction by PLE in an Intermittent Process

Preliminary tests were carried out to choose the solvent to be used in PLE. In this study, the process variables were kept fixed at 144 °C, 4 extraction cycles, and 90% rinse volume.

The chosen solvents were mainly based on factors that highlight sustainability and health safety (GRAS solvents—Generally Recognized as Safe). Thus, water, anhydrous ethanol (99%) and hydrated ethanol (40%) were chosen as solvents.

The extracts were produced using the Pressurized Liquid Extraction (PLE) intermittent process, using the equipment Dionex ASE 150 (Thermo Scientific, Sunnyvale, CA, USA) and ethanol 40% (*v*/*v*) as pressurized solvent. Briefly, approximately 5 g of triturated grape by-product (wet or dried in an oven at 105 °C for 36 h) was transferred to an extraction cell (34 mL capacity) containing a cellulose filter on the bottom. The cell was placed on the PLE equipment, where the extraction occurred at 10.35 MPa.

A Central Composite Rotational Design (CCRD) was used to study the extraction of extract rich in phenolic compounds and pigments from grape by-product. Process parameters such as temperature (T, °C), number of extraction cycle (N) and rinse volume (RV, % of cell capacity) were selected as independent variables (Table 1). The contact time between the solvent (hydrated ethanol 40%) and the matrix in each cycle, static time (St), was kept constant (5 min) throughout the process.

The total phenolic content (TPC), the anthocyanin content and antioxidant activity of the extract were selected as the response variables.

To estimate the effect of independent variables on the extraction of total phenolic compounds, main effects and variance analyses (ANOVA) were performed. The best condition was evaluated by response surface analysis or contour curves, depending on the TPC and anthocyanin contents. All statistical treatment was applied to experimental data using Statistica^®^ 10 software.

Models were adjusted to experimental data and their predictive capabilities were also evaluated by ANOVA.

### 2.3. Production and Characterization of Grape Pomace Extract

The best extraction parameters defined by the experimental design (CCRD) were applied to prepare grape pomace extract using pressurized ethanol 40% (*v*/*v*) as solvent. The extract was produced in triplicate and evaluated regarding the total phenolic compounds content, total anthocyanins, and antioxidant capacity.

#### 2.3.1. Determination of Total Phenolic Content (TPC)

For TPC, analysis was carried out following the Folin–Ciocalteu colorimetric method proposed by Singleton and Rossi (1965), with modifications [22]. Briefly, 500 µL of diluted extract was mixed with 2.5 mL Folin–Ciocalteu aqueous solution (1:10 *v*/*v*) in a test tube. After 5 min, 2.0 mL of sodium carbonate aqueous solution (4%) was added to the tube, which was kept in absence of light during 45 min for color development. Then, the sample was evaluated using a spectrophotometer (Genesys 10S, Thermo Scientific, USA) at 760 nm wavelength and the TPC was calculated using a standard gallic acid curve (y = 132.0076x + 0.0266, R^2^ = 0.9984).

#### 2.3.2. Determination of Total Anthocyanins (TA)

The TA content of the extract was evaluated according to Lee et al. (2005) [23]. Firstly, the extract was diluted 100-fold in potassium chloride buffer (0.025 M, pH 1) and sodium acetate buffer (0.4 M, pH 4.5). Then, the absorbance of the dilutions was recorded at 520 and 700 nm using a Spectrophotometer (Genesys 10S, Thermo Scientific, USA). The TA was calculated using Equation (1):TA = (A × MW × DF × 1000)/ε(1)
where A is {[(Absorbance 520 nm − Absorbance 700 nm) pH1] − [(Absorbance 520 nm − Absorbance 700 nm) pH 4.5]}; MW is the molecular weight of yanidin-3-O-glucoside (449.2 g/mol); DF is the dilution factor; and ε is the molar extinction coefficient for cyanidin-3-O-glucoside (26,900 L/mol cm).

#### 2.3.3. Determination of Antioxidant Capacity by FRAP

The antioxidant capacity of the extracts was evaluated in terms of Ferric Reducing Antioxidant Power (FRAP), according to Pulido, Bravo and Saura-Calixto (2000) [24]. For this purpose, 90 μL of diluted extracted was transferred to test tubes and added by 270 μL of distilled water and 2.7 mL of the FRAP reagent. The tubes were shaken in a thermal bath (TE-053, Tecnal, Piracicaba, SP, Brazil) at 37 °C for 30 min. Then, the absorbance of the samples was recorded at 595 nm (Genesys 10S, Thermo Scientific, USA) and the antioxidant capacity was calculated using a standard FeSO_4_ curve (y = 28.917x − 0.020, R^2^ = 0.988).

### 2.4. Microencapsulation by Spray-Drying

Grape pomace extract produced using the best extraction parameters was microencapsulated using a pilot scale spray-dryer (MSD 1.0, Lab Maq do Brasil Ltda, Ribeirão Preto, SP, Brazil). First, the grape pomace extract was concentrated 2-fold using a rotary evaporator operating at 40 °C and under vacuum atmosphere. Next, the concentrated extract was mixed with maltodextrin (90:10 *v*/*w*) by magnetic stirring at room temperature. Afterwards, the mixture was atomized in the spray dryer using a 2 mm nozzle and feed rate of 0.6 L.h^−1^. The process conditions used were the following: inlet air velocity of 2.5 m.s^−1^; inlet temperature of 140 °C; air compressor pressure of 0.2 MPa. Particles were produced in triplicate for characterization analysis.

### 2.5. Microparticles Characterization

#### 2.5.1. Phytochemicals and Antioxidant Activity

For the extraction of phytochemical compounds, 0.5 g of particles was transferred to a falcon tube. Then, 10 mL of ethanol 40% was added and mixed on a vortex shaker for 2 min (this process was performed twice). The mixture was then transferred to a beaker and placed on an ultrasound bath for 15 min. The sample was centrifuged for 5 min at 3000 rpm and 25 °C (Centrigure 5430R, Eppendorf, Germany). The supernatant was collected. TPC and TA were carried out according to methodologies described above in Section 2.3. The recovery yield was calculated by the percentage of phytochemicals detected in the collected powder in relation to the theoretical amount. The latter parameter was calculated by considering the concentration of phytochemicals in the extract and the volume of extract subjected to spray-drying.

#### 2.5.2. Particle Size, Distribution, and Morphology

For size distribution evaluation, particles were dispersed in ethanol under an ultrasonic bath for 30 s for the disruption of possible aggregates. Then, the dispersion was evaluated in a laser diffraction equipment (SALD 201/V, Shimadzu, Kyoto, Japan) working in moderate pressure. For this analysis, the dried sample was fixed onto a carbon tape and analyzed without any treatment [25].

The morphology of particles was accessed by Scanning Electron Microscopy (TM 3000 Tabletop, Hitachi, Tokyo, Japan). For this analysis, the dried sample was fixed onto a carbon tape. Images were acquired at magnifications of 1.500 and 3.000×, with an electron beam acceleration of 5.0 kV.

#### 2.5.3. Fourier-Transformed Infrared Spectroscopy (FTIR)

The chemical structures of the produced microparticles, lyophilized grape by-product extract, and maltodextrin were evaluated by FTIR equipment (Spectrum One, PerkinElmer, Norwalk, CA, USA), equipped with an attenuated total reflection (ATR) accessory. The spectra were recorded in transmittance mode in a spectral region of 4000–600 cm^−1^.

## 3. Results and Discussion

### 3.1. Preliminary Tests to Choose the Extraction Solvent

Hydrated ethanol (40%) positively influenced the increase in compounds of interest in the extract composition (Figure 1). The type of solvent and its polarity can affect the transfer of electrons and hydrogen atoms, which is also a key aspect in measuring antioxidant capacity. Ethanol and water are two polar solvents that are frequently used for the extraction of bioactive compounds from food matrixes. In general, its polar character results in affinity with phenolic compounds. The ethanol–water mixture was much more efficient for the extraction of phenolic compounds, either for wet (97.9 ± 8.3 mg GAE/g, d.b.) or dried grape pomaces (95.2 ± 7.0 mg GAE/g, d.b.) (Figure 1A).

For the total anthocyanin content, anhydrous ethanol (99%) demonstrated itself to be a better solvent for this process (Figure 1B). Drying the grape pomace does not positively interfere with the extraction of bioactive compounds (Figure 1). Thus, the application of wet grape pomace, in addition to favoring the extraction of the bioactive, also favors the process in economic terms. Given this, the drying process may be applied considering the amount of by-product to be processed, the available time, and other economic aspects.

Concerning these results, hydrated ethanol (40%) was chosen as solvent for the experimental design applied to study the bioactive extraction of wet grape pomace.

### 3.2. PLE Extraction Conditions Using Grape Pomace

Table 2 shows the results of CCRD 2^3^ performed to evaluate how the extraction parameters, Temperature (°C) (X1), number of cycles (N) (X2) and the rinse volume (RV) (% of cell capacity) (X3), affected the responses of TPC (mg GAE/g, d.b.), anthocyanins (mg/g, d.b.) and antioxidant activity (FRAP, μmol FeSO_4_/g, d.b.).

The results show that the wet grape pomace extract presents high levels of total phenolic content, which ranged from 15.7 to 97.4 mg GAE/g and also antioxidant activity, from 2.58 to 9.95 μmol FeSO_4_/g (Table 2).

In the statistical analysis of the main effects of process variables, temperature was the only significant variable in the process, indicating that the number of cycles (directly related to extraction time) and rinse volume did not influence the obtaining of extracts rich in pigments and compounds (Figure 2). These are great advantages of PLE in an intermittent process. Also, due to the quick extraction time, the use of high temperature does not degrade the active compounds.

The volume of solvent used in rinsing (RV) varied from 40 to 140% of the extraction cell volume (34 mL), therefore varying from 13.6 to 47.6 mL. Since the volume of solvent is not a significant variable, instead of using 47.6 mL as a rinse volume, 13.6 mL is sufficient, which represents a 100% savings, ensuring that this low volume can be used for obtaining extract with the same bioactive compounds content.

For some vegetal matrices, the number of cycles (N), which represents how many times the rinse solvent will be in contact with the matrix in the intermittent PLE process, influences the extraction process, but for grape pomace, this behaviour was not observed (Figure 2).

In order to achieve a model that fits the behaviour of the experimental data, the linear and quadratic model coefficients were generated and analysed by ANOVA and the results are shown in Table 3.

Although it is not shown in the Pareto diagram (Figure 2) in the model coefficients (Table 3), it can be seen that there was no significant interaction between the process variables, between T vs. N nor T vs. RV.

The first-order model was better to predict total anthocyanins (TA) content (R^2^ = 0.94) than the second-order model (R^2^ = 0.81). The same tendency was observed for the antioxidant activity; even if the models were not significant (*p* > 0.05). As expected, the significant coefficients in the model were only those related to temperature (T), which had already been demonstrated in the main-effect analysis (Figure 2).

The adjusted linear model enabled us to generate the contour curve that shows the influence of the variables T vs. N and T vs. RV on the total anthocyanin content (mg/g, d.b.). For lower temperature conditions, below 85 °C, the highest anthocyanin levels were obtained, regardless the number of cycles (N) and rinse volume (RV). Note that for the entire range of values studied (N and RV), the total anthocyanin content is practically the same (Figure 3A,B).

Concerning the phenols (TPC), the second-order model presented higher values of the regression coefficient (R^2^ = 0.96), superior to the first-order model (R^2^ = 0.91). For this answer, the linear coefficient of the variable that represents the number of cycles (N) in the quadratic model was also significant (β_2_, Table 3) as was the quadratic coefficient of temperature (β_11_, Table 3). The quadratic model, predictive of this experimental behavior, showed that both the temperature (T) and the number of cycles (N) influenced the content of phenolic compounds in the extracts. Higher activities were verified in extracts obtained at high temperatures (Figure 4).

In general, the temperature for optimal extraction differs between the compounds being extracted. In fact, our results show that the maximum recovery of TPC took place when the highest temperature was applied (144 °C) (97.4 ± 1.1 mg GAE/g, d.b.) (Figure 4). At high temperatures, the vegetal tissue is softened, and, thus, the linkages between phenolic compounds and other matrix components (polysaccharides, proteins, and inorganic compounds) are weakened. This enhances the mass transfer of phenolic compounds to the extract, resulting in a higher yield [26]; however, such a high temperature impacted the anthocyanins from grape by-product. Obviously, the heat stability of anthocyanins depends on several aspects, such as the raw starting material, pH, and co-pigments. Also, some anthocyanins are more susceptible to heat than others [27]. In this way, the results suggest that the extracted anthocyanins from grape pomace present less heat stability, achieving the highest yield when the extraction temperature was between 40 and 85 °C (3.99 ± 0.07 mg/g, d.b.) (Table 2 and Figure 3).

A similar behavior for anthocyanins and TPC was verified by Ju and Howard [28] while preparing dried grape skin extracts in PLE. In their study, it was verified that high temperatures favored the extraction of phenolic compounds, while disfavoring the total anthocyanin content in the extracts. Moreover, these authors used different extraction solvents (acidified water and acidified methanol) and they verified that the behavior was the same, regardless of the extraction solvent used.

Despite the low total anthocyanin content, the FRAP value of the extract obtained at the highest temperature was one of the highest among the produced samples, indicating that the phenolic compounds extracted have good antioxidant activity, and thus, can be indicated for application as high-value added ingredient. Therefore, the sample 10 from the experimental design (Table 2) was encapsulated by spray-drying due to its higher values of TPC (97.4 ± 1.1 mg GAE/g, d.b.) and FRAP (7.14 ± 0.46 μmol FeSO_4_/g, d.b.).

### 3.3. Powders Characterization

#### 3.3.1. Mean Diameter Size and Distribution, Morphology of the Particles and Phenolic Compounds

The produced particles presented a narrow and monomodal distribution (Span = 1.6) (Figure 5). The mean particle size was 7.73 ± 0.95 μm and most of the particles were smaller than 17 μm (D_90_ = 16.49 μm).

The final product appeared as a white powder which was analyzed by Scanning Electron Microscopy (SEM), showing the particles in Figure 6, in agreement with the results provided by size distribution analysis. Regarding their morphology, particles had a nearly spherical shape with some irregular depression at the surface.

The occurrence of these depressions is somehow normally observed for particles produced using the spray-drying technique, which is usually attributed to the viscoelastic properties of the wall material and to the solvent evaporation rate during the drying process [19,29]. In fact, it had already been demonstrated that the lower the solvent evaporation rate, the higher the occurrence of depression on particles prepared using maltodextrin DE10 as wall material. This occurs because slow solvent evaporation causes a slow hardening of particle crust, which enables the formation of shrivel points [13]. Even though the particles presented depressions at their surface, no cracks were observed at SEM images, indicating that particles were physically intact. Similar results were also described before for particles containing grape polyphenols encapsulated through the spray-drying technique [30,31].

While the extract and maltodextrin formed a solution in the feed, they also formed continuous and homogeneous walls of the particles, which were possibly hollow, because of the prompt solvent evaporation from the interior to the surface of the droplets. The particles thus constituted were of the matrix-type, so they can also be called microspheres, but not microcapsules.

The particles presented 38.1 ± 3.6 mg GAE/g of particles and 1.40 ± 0.22 mg TA/g of particles. The recovery yield for TPC was 79%, suggesting that phytochemicals were minimally affected by the temperature used in the drying process (140 °C). In short, the spray-dryer’s operational conditions used for dehydrating the grape pomace extract and the use of maltodextrin as carrier were efficient in the encapsulation and hence the preservation of its bioactive compounds. In addition, according to Mesquita et al. (2023) [32], besides the protective effect on the compounds during the drying and storage of the powders, the presence of maltodextrin improves the quality of the powders obtained by providing increased dispersibility and reducing hygroscopicity, agglomeration and mean particle diameter.

However, the stability along the storage as well as bioaccessibility and transepithelial transport of these bioactive compounds should be investigated in a future study.

#### 3.3.2. Chemical Characterization by FTIR

FTIR spectra of the produced particles, as well as maltodextrin and lyophilized grape pomace extract are presented in Figure 7.

Some important peaks are highlighted in Figure 7 for each sample evaluated. The peaks at 3305 cm^−1^, 1724 cm^−1^ and 1029 cm^−1^ in the grape pomace extract spectra are, respectively, related with the vibration of OH groups of phenolics and sugars, the C=O stretching of hydrolysable esters and tannins, and the C-O stretching of methoxy groups (i.e., alcohol, ester, and ether). All of them are common chemical structures to the phytochemicals present in this type of material [31]. Comparatively in the infrared spectra of maltodextrin, the occurrence of a peak at 1015 cm^−1^ and a higher peak spare at 1080 cm^−1^, which refer to the stretching of C-O and angular strain of =CH and =CH2, respectively, are in turn characteristic chemical structures of carbohydrates [15]. Interestingly, the infrared spectra of the microparticles produced in this work present a format very similar to that of the maltodextrin, except for a change in the peak at 1080 cm^−1^, which may be related to the presence of extract on the surface of the particles. We can therefore infer that maltodextrin, in fact, plays the role of englobing the grape pomace extract, being present in an expressive way outside the particle.

## 4. Conclusions

Developing green processes for further exploring the agro-industrial residues is an important tool to comply with the circular economy in the food sector. In this work, process parameters for the production of grape pomace rich-phenolics and anthocyanins extracts through the PLE technique have been established. The type of solvent applied was of utmost importance for the recovery of bioactive compounds from grape pomace by PLE. Ethanol 40% was the most effective solvent for TPC extraction, whereas ethanol 99% was more effective for AT, regardless the condition of grape pomace (wet or dried using mild conditions).

A CCRD was used to study the extraction process, in which the temperature, number of extraction cycles, and rinse volume were selected as independent variables. The first-order model was better to predict anthocyanins, and the second-order model was more indicated for TPC. The temperature was shown to be an important variable in the extraction process, in which the higher the temperature, the higher the phenolic compound concentration in the extract. The recovery of phenolic compounds from grape pomace reveals the importance of this study on adding value to a waste from the wine making process and encourages circular economy and sustainable development.

The encapsulation of grape pomace rich-phenolics and anthocyanin extract by spray-drying using maltodextrin as wall material proved to be an efficient technique for protecting these compounds during the dehydration process. In general, the applied process of encapsulation also produces powdered extracts that are microbial and chemically more stable than aqueous extracts. These powders also demand less space for packing, transportation, and storage than the liquid extracts, which reduce costs and facilitate their commercialization.

Overall, the results presented in this research show that the PLE extraction followed by extract encapsulation by spray-drying using maltodextrin as carrier represents an advantageous alternative for the valorization of grape pomace.

## Figures and Tables

**Figure 1 foods-13-00279-f001:**
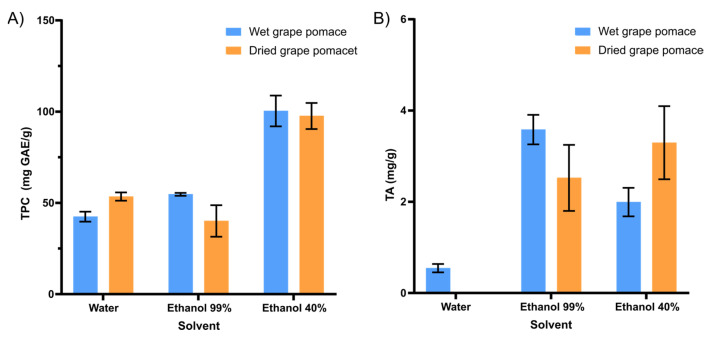
Influence of the type of solvent (water, ethanol 99%, ethanol 40%) and drying of the raw material on (**A**) the content of Total Phenolic Compounds (TPC, mg GAE/g, d.b.) and (**B**) Total Anthocyanins (TA, mg/g, d.b.) in grape pomace extracts.

**Figure 2 foods-13-00279-f002:**
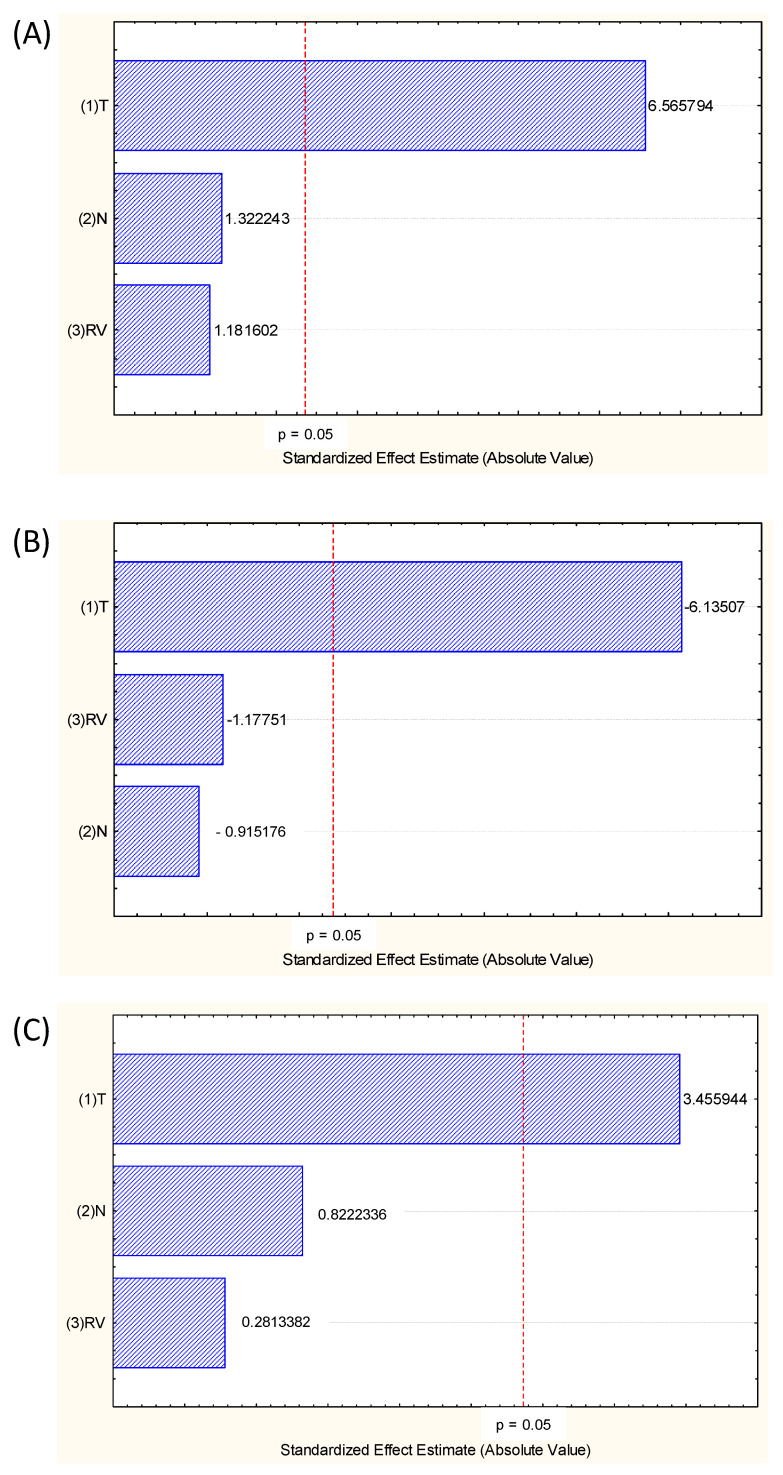
Pareto chart showing the main effects of the variables Temperature (°C) (X1), number of cycles (N) (X2) and rinse volume (RV) (%) (X3) for the three responses analyzed, (**A**) TPC (mg GAE/g BP, d.b.), (**B**) total anthocyanins (mg/g BP, d.b.) and (**C**) antioxidant activity (FRAP, μmol FeSO_4_/g BP, d.b.).

**Figure 3 foods-13-00279-f003:**
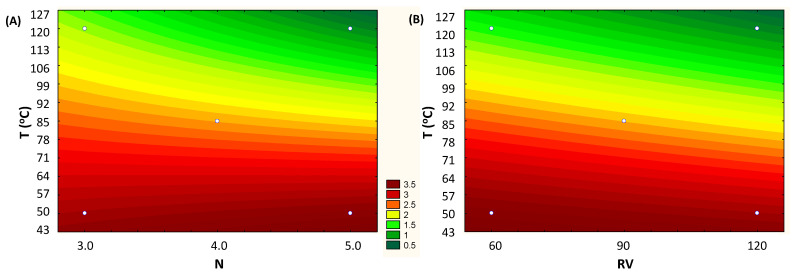
Fitted contour curves from the first-order models for the response concentration of Total Anthocyanins (mg/g, d.b.), according to the independent variables: (**A**) T (Temperature, °C) vs. N (number of cycles) and (**B**) T (Temperature, °C) vs. RV (rinse volume, % of cell capacity).

**Figure 4 foods-13-00279-f004:**
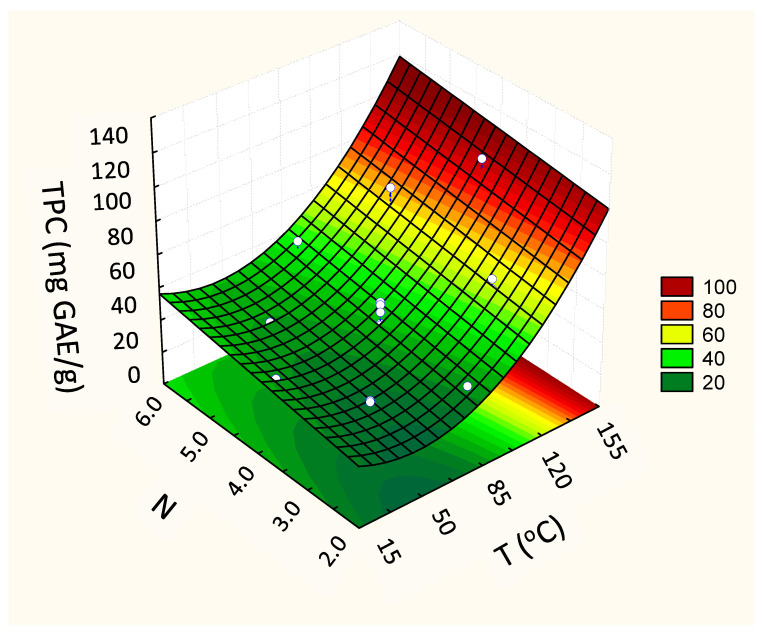
Fitted response surface from the second-order models for the response Total Phenolic Content (TPC) (mg GAE/g, d.b.), according to the independent variables TPC vs. T (Temperature, °C) and N (number of cycles).

**Figure 5 foods-13-00279-f005:**
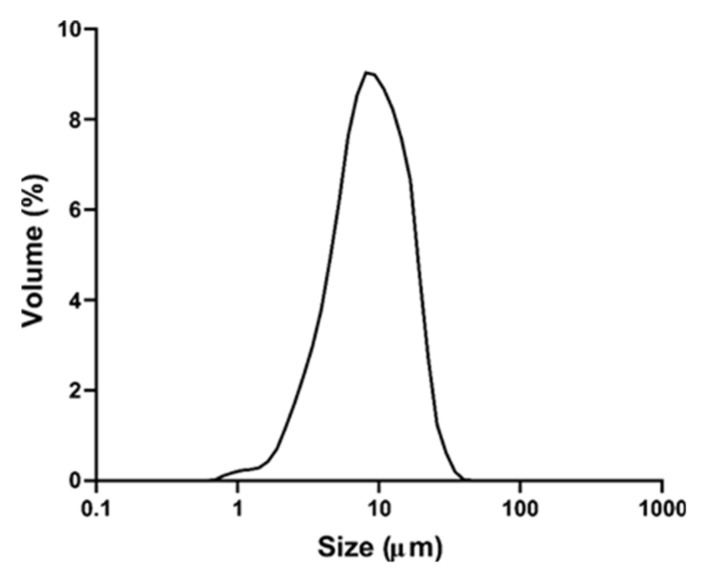
Size distribution of particles containing TPC from grape by-product.

**Figure 6 foods-13-00279-f006:**
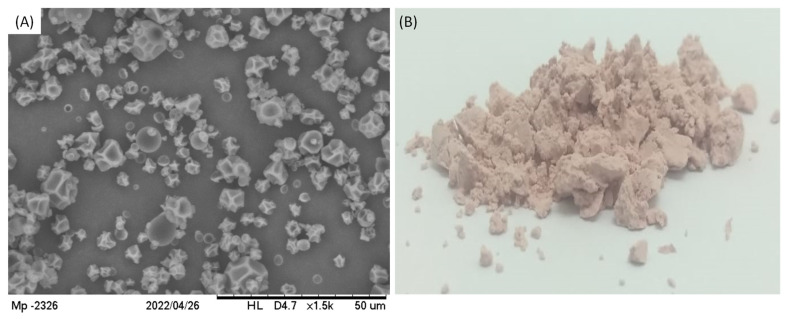
(**A**) SEM images of particles containing TPC from grape pomace at magnification of 1500× and (**B**) the image of the powder product.

**Figure 7 foods-13-00279-f007:**
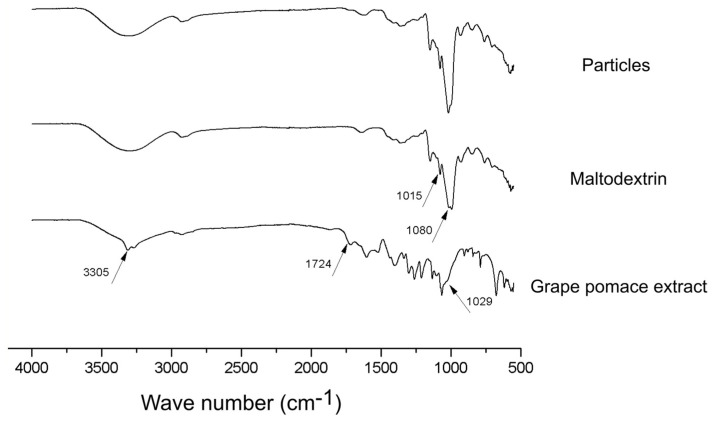
FTIR spectra of the produced particles, maltodextrin and grape pomace extract.

**Table 1 foods-13-00279-t001:** Independent variable levels in the CCRD applied to study the extraction process of phenolic compounds and pigments from grape by-product. Temperature (T) (X_1_), number of extraction cycles (N) (X_2_) and rinse volume (RV) (X_3_) as independent variables.

Independent Variable\Coded Values →	−1.68	−1	0	1	1.68
T (X_1_) (°C)	26	50	85	120	144
N (X_2_) (n)	2	3	4	5	6
RV (X_3_) (%)	40	60	90	120	140

**Table 2 foods-13-00279-t002:** Total Phenolic Content (TPC, mg GAE/g, d.b.), total anthocyanins (mg/g, d.b.) and antioxidant activity by FRAP (μmol FeSO_4_/g, d.b.) for the tests performed according to CCDR 2^3^ of extract. The variables were Temperature (°C) (X1), number of cycles (N) (X2) and rinse volume (RV) (% of cell capacity) (X3) for the three responses analyzed, (A) TPC (mg GAE/g BP, d.b.), (B) total anthocyanins (mg/g BP, d.b.) and (C) antioxidant activity (FRAP, μmol FeSO_4_/g BP, d.b.).

Sample	T (°C)	N	RV (%)	TPC (mg GAE/g, d.b.)	Anthocyanins (mg/g, d.b.)	FRAP (μmol FeSO_4_/g, d.b.)
1	50 (−1)	3 (−1.0)	60 (−1)	16.4 ± 0.2	3.4 ± 0.09	2.81 ± 0.22
2	120 (+1)	3 (−1.0)	60 (−1)	48.9 ± 0.2	1.65 ± 0.04	6.40 ± 0.43
3	50 (−1)	5 (+1.0)	60 (−1)	21.5 ± 2.1	3.66 ± 0.09	3.04 ± 0.22
4	120 (+1)	5 (+1.0)	60 (−1)	51.5 ± 1.3	1.59 ± 0.01	6.01 ± 0.97
5	50 (−1)	3 (−1.0)	120 (+1)	15.7 ± 0.1	3.21 ± 0.10	2.58 ± 0.04
6	120 (+1)	3 (−1.0)	120 (+1)	55.6 ± 0.3	1.87 ± 0.04	5.46 ± 0.41
7	50 (−1)	5 (+1.0)	120 (+1)	22.2 ± 0.1	3.38 ± 0.15	3.89 ± 0.76
8	120 (+1)	5 (+1.0)	120 (+1)	72.1 ± 1.5	0.26 ± 0.08	7.39 ± 0.41
9	26 (−1.68)	4 (0)	90 (0)	20.5 ± 0.2	3.44 ± 0.04	3.64 ± 0.16
10	144 (+1.68)	4 (0)	90 (0)	97.4 ± 1.1	1.06 ± 0.58	7.14 ± 0.46
11	85 (0)	2 (−1.68)	90 (0)	21.9 ± 0.4	2.61 ± 0.13	4.74 ± 0.57
12	85 (0)	6 (+1.68)	90 (0)	41.7 ± 0.2	3.99 ± 0.07	6.43 ± 0.12
13	85 (0)	4 (0)	40 (−1.68)	38.1 ± 0.4	3.38 ± 0.06	5.73 ± 0.53
14	85 (0)	4 (0)	140 (+1.68)	35.8 ± 1.3	3.44 ± 0.15	9.95 ± 1.57
15	85 (0)	4 (0)	90 (0)	54.6 ± 0.7	2.28 ± 0.01	4.29 ± 0.75
16	85 (0)	4 (0)	90 (0)	23.4 ± 0.5	2.21 ± 0.06	3.66 ± 0.12
17	85 (0)	4 (0)	90 (0)	32.1 ± 0.2	1.70 ± 0.50	7.80 ± 0.50

Data presented as mean ± standard deviation.

**Table 3 foods-13-00279-t003:** Coefficient of regression of process variables and analysis of variance (ANOVA) for first and second-order models for the response variables: concentration of Total Phenolic Content (TPC, mg GAE/g BP, d.b.), (B) Total Anthocyanins (mg/g BP, d.b.) and (C) Antioxidant Activity (FRAP, μmol FeSO_4_/g BP, d.b.).

	TPC (mg GAE/g, d.b.) (Y1)	Anthocyanins (mg/g, d.b.) (Y2)	Antioxidant Activity (FRAP, μmol FeSO_4_/g, d.b.) (Y3)
First-order model			
β_0_	34.9 *	2.29 *	4.85 *
Linear Coefficient			
β_1_	19.0 *	−1.04 *	1.62 *
β_2_	3.83	−0.155	0.38
β_3_	3.42	−0.199	0.13
Interaction			
β_12_	0.947	−0.261	0.001
β_13_	3.41	−0.081	−0.022
β_23_	1.91	−0.205	0.423
R^2^	0.91	0.94	0.69
F_calculated_	6.33	10.0	1.46
F_tabulated_	6.16	6.16	6.16
*p*	0.048	0.022	0.373
Second-order model			
β_0_	27.1 *	2.11 *	
Linear Coefficient			
β_1_	20.6 *	−0.902 *	
β_2_	4.69 *	−0.079	
β_3_	1.71	−0.110	
Quadratic Coef.			
β_11_	10.2 *	−0.083	
β_22_	0.556	0.288	
β_33_	2.38	0.327	
Interaction			
β_12_	0.947	−0.261	
β_13_	3.41	−0.081	
β_23_	1.91	−0.205	
R^2^	0.96	0.81	
F_calculated_	59.8	24.0	
F_tabulated1 (*p* = 0.05)_	6.94	4.54	
F_lack of fit_	7.56	5.15	
F_tabulated2 (*p* = 0.05)_	4.67	19.41	

F_calculated_ > F_tabulated1_: a significant model; F_Lack of fit_ < F_tabulated2_: a significant and predictive model. * Indicates significance at 95% confidence interval (*p* < 0.05). Index 1 refers to the Temperature; index 2 refers to the number of cycles, and 3 refers to the rinse volume (X3).

## Data Availability

Data are contained within the article.

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
