# Peer review of "Grape Pomace Rich-Phenolics and Anthocyanins Extract: Production by Pressurized Liquid Extraction in Intermittent Process and Encapsulation by Spray-Drying"

_foods, 2024, doi:10.3390/foods13020279_

Round 1

Reviewer 1 Report

Comments and Suggestions for Authors

In the present study, the authors propose the standardization of polyphenol extraction from grape pomace using PLE (Pressurized Liquid Extraction). The work lacks novelty as it relies on a widely used technique, and the selected matrix, grape pomace, has been extensively studied. Moreover, the conducted assays are superficial. The authors should have included a study of the compound profile. It is well-known that the Folin technique, although rapid, is a colorimetric method prone to many interferences. While I understand its use for standardization, after selecting certain conditions, the authors should have performed a comprehensive compound profile study. Whether the total compound content is determined to be higher or lower by FOLIN likely does not reflect what is observed in the polyphenol profile, including anthocyanins. The authors did not use acid, a common practice in extraction when dealing with a matrix rich in anthocyanins. The difference observed in Total Phenolic Content (TPC) and anthocyanins is possibly due to the lack of acid utilization, which may not affect polyphenol extraction but can impact anthocyanin extraction yield. This is crucial, especially considering the well-studied nature of red grape pomace.

On another note, they encapsulate the extract with a matrix containing only MD (highly soluble) without comparing the yields with other encapsulating matrices. While they characterize the microcapsules, they do so at time zero without considering the protection, or lack thereof, provided by encapsulation over time. I understand that the FTIR assays aim to demonstrate the effective encapsulation of polyphenols, but the authors could have extracted surface polyphenols and subtracted them from the total, providing a value for the polyphenols effectively encapsulated.

Author Response

We appreciate the referee’s comments on our manuscript and we are sorry to hear that the referee’s expectations were not fulfilled.

It is important to clarify that the main purpose of this work was to offer an alternative for the valorization of grape pomace, and thus, to comply with the circular economy in the food industry. This imply that our main goal was to recovery the higher possible amount of bioactive compounds from our wine making waste (grape pomace) and not necessarily to study the extraction process itself.

Thus, the process parameters for PLE extraction were well explored and described in our work, as they should be. The extract encapsulation was a secondary aim here, and thus, the spray-drying process was not extensively explored as expected by the referee.

We would like to stress that the data presented in this manuscript are being used to developed a second phase of our study, at which the extraction and encapsulation were scaled-up. The obtained particles will be carefully characterized and their use will be well-described in a food application. We hope to publish these new data soon.

Concerning the total anthocyanin contents, we´ve been used a standard method (Lee et al. (2005), in which the total anthocyanin pigment content (of fruit juices, beverages, natural colorants, and wines) are determined by the pH differential method. We agree with the reviewer´s observation about the acid extraction and this modification will be carried out in our future works, as well as the polyphenol profile characterization.

Reviewer 2 Report

Comments and Suggestions for Authors

Change the contour plot-   Fitted contour curves from the first order models for the response concentration of Total Anthocyanins (mg/g, d.b.), accord-317 ing to the independent variables: (A) T (Temperature) vs N (number of cycles (N) and T (Temperature) vs RV (rinse volume).  Replace it with another type of graph

Present SEM micrographs at the same scale. 

Modifying the scale of figure 4. cannot be interpreted .

Figure 4. Fitted response surface from the second order models for the response Total Phenolic Content (TPC) (mg GAE/g, d.b.), 336 according to the independent variables T (Temperature) vs RV (rinse volume).

Clearly state the findings in the summary 

Restructure the conclusions 

Author Response

Dear Reviewer,

We greatly appreciate your comments, which helped us to improve the manuscript. Please, read below our answers. All the changes made in the manuscript are shown in red in the revised manuscript (R1).

Sincerely yours,

Milena Martelli Tosi

Change the contour plot-   Fitted contour curves from the first order models for the response concentration of Total Anthocyanins (mg/g, d.b.), according to the independent variables: (A) T (Temperature) vs N (number of cycles (N) and T (Temperature) vs RV (rinse volume).  Replace it with another type of graph

A: We changed the figure legend, thanks to your comments, and changed the coded values to real values in the figure axis. Nevertheless, we preferred to maintain the same type of graphics because it is common in similar articles.

Present SEM micrographs at the same scale.

A: Thank you for this observation. We kept only one SEM image to prevent errors and included the picture of the particles.

Modifying the scale of figure 4. cannot be interpreted .

Figure 4. Fitted response surface from the second order models for the response Total Phenolic Content (TPC) (mg GAE/g, d.b.), according to the independent variables T (Temperature) vs RV (rinse volume).

A: Figure 4 was replaced, and we updated its legend accordingly.

Clearly state the findings in the summary

A: The summary has been rewritten.

Restructure the conclusions

A: The conclusion has been restructured.

We appreciate the referee’s comments on our manuscript. All the suggested changes were incorporated in the new version of the manuscript.

Reviewer 3 Report

Comments and Suggestions for Authors

The paper entitled “Grape pomace rich-phenolics and anthocyanins extract: production by pressurized liquid extraction in intermittent process and encapsulation by spray-drying” is authored by Jéssica Thaís do Prado Silva, Millene Henrique Borges, Carlos Antonio Cardoso De Souza, Carmen Silvia Favaro-Trindade, Paulo José do Amaral Sobral, Alessandra Lopes de Oliveira and Milena Martelli-Tosi, presents the results of encapsulation of bioactive compounds by using the spray-drying process and it was concluded that 40% ethanol was most effective for obtaining extracts with higher total phenolic contents. The authors concluded that  the obtained results showed that extraction followed by encapsulation of grape pomace extract is a good strategy to simplify future applications, whether for food, cosmetics or pharmaceutical fields. The authors cited relevant literature, thoroughly described the applied methods and provided all needed information for repetition of the experiments. In my opinion, the paper is nicely organized, well written, however it needs to be revised based on the points below.

1.       Introduction: line 95 – please specify the characterization methods of the obtained microparticles;

2.       In the subsection 2.2. the validation approaches of the models should be emphasized;

3.       Table 2. – what the values after ± sign represent?

4.       More validation parameters must be provided and discussed for the established models. The R2 does not necessarily mean that a model predicts well.

5.       Line 308: R2 should be instead of R2. It is determination coefficient.

6.       Figure 4 is unreadable and should be replaced.

7.       Some introductory sentences (max. 2) are missing in Conclusion section.

Comments on the Quality of English Language

There are some minor mistakes that should be corrected.

Author Response

Dear Reviewer,

We greatly appreciate your comments, which helped us to improve the manuscript. Please, read below our answers. All the changes made in the manuscript are shown in red in the revised manuscript (R1).

Sincerely yours,

Milena Martelli Tosi

  1. Introduction: line 95 – please specify the characterization methods of the obtained microparticles;

A: Done.

  1. In the subsection 2.2. the validation approaches of the models should be emphasized;

A: As our primary goal was to optimize both anthocyanin and TPC contents, which showed differences in the model results, we opted not to conduct a model validation study. Instead, we selected the best condition from the experimental design. This was clarified in the text:

Our results show that the maximum recovery of TPC took place when the highest temperature was applied (144 oC) (97.4 ± 1.1 mg GAE/g, d.b.) (Figure 4). However, this elevated temperature adversely affected the anthocyanins in the grape by-product. Consequently, we chose the best conditions using response surface analysis or contour curves, depending on the TPC and anthocyanin contents. Sample 10 from the experimental design (Table 2) was selected for encapsulation due to its higher TPC (97.4 ± 1.1 mg GAE/g, d.b.) and FRAP (7,138 ± 460 mmol FeSO4/g, d.b.) results.

  1. Table 2. – what the values after ± sign represent?

A: These values represent the standard deviation among the evaluated triplicates. This information is now reported as foot note in Table 2.

  1. More validation parameters must be provided and discussed for the established models. The R2 does not necessarily mean that a model predicts well.

A: We agree with the reviewer; nevertheless, our response variables (TPC and anthocyanins) exhibited antagonistic results, as previously explained.

  1. Line 308: R2 should be instead of R2. It is determination coefficient.

A: Done.

  1. Figure 4 is unreadable and should be replaced.

A: Figure 4 was replaced.

  1. Some introductory sentences (max. 2) are missing in Conclusion section.

A: The conclusion has been rewritten, and the introductory sentences were added.

Comments on the Quality of English Language

There are some minor mistakes that should be corrected.

A: We reviewed the manuscript and some English mistakes were corrected.

We appreciate the referee’s comments on our manuscript. All the suggested changes were incorporated in the new version of the manuscript.

Round 2

Reviewer 1 Report

Comments and Suggestions for Authors

Even though authors have explained the objetive of their study, they did not answear my concers. Furthermore, they estblished that will evaluate the use of acid in further studies, but they should have carried it out in this first study.

The authors have not made any changes that would allow for a reevaluation of the work, and therefore, I believe it should be rejected.
